# Validity of Heart Rate Variability Measured with Apple Watch Series 6 Compared to Laboratory Measures

**DOI:** 10.3390/s25082380

**Published:** 2025-04-09

**Authors:** Lauren Bonneval, David Wing, Sydney Sharp, Maira Tristao Parra, Ryan Moran, Andrea LaCroix, Job Godino

**Affiliations:** 1San Diego General Preventive Medicine Residency, Herbert Wertheim School of Public Health and Human Longevity Science, University of California San Diego, La Jolla, CA 92093, USA; lbonneval@health.ucsd.edu (L.B.);; 2Exercise and Physical Activity Resource Center, Herbert Wertheim School of Public Health and Human Longevity Science, University of California San Diego, La Jolla, CA 92093, USA; dwing@health.ucsd.edu (D.W.); sydneys@fhcsd.org (S.S.); alacroix@health.ucsd.edu (A.L.); 3Family Health Centers of San Diego, San Diego, CA 92102, USA; 4International Consulting Associates Inc., Arlington, VA 22201, USA

**Keywords:** heart rate variability (HRV), Apple Watch, cardiovascular disease (CVD)

## Abstract

We assessed the test validity of the Apple Watch series 6 measure of heart rate variability (HRV) by comparing it with the reference measure assessed via a Biopac 3-lead electrocardiogram (ECG). We recruited 78 healthy adults (aged 20–75 years). HRV was measured using an in-lab protocol while resting, talking, watching a movie, before walking, and after walking. We conducted a synchronized countdown for each condition to guarantee that the recordings would be aligned between the two devices by using event markers in the Biopac at the exact time that the Apple Watch Breathe app began and ended. We assessed test validity using the Bland–Altman method, and both precision and accuracy were estimated using Lin’s concordance correlation coefficient. The highest level of agreement and concordance between devices occurred during rest. We observed near-perfect agreement for R-R intervals and beats per minute (BPM) measures, with mean absolute percentage errors (MAPE) of 1.15% during resting conditions. We observed moderate levels of agreement and concordance for N-N intervals at rest with a MAPE of 31.31% during resting conditions. The Apple Watch provides a high level of validity for measuring R-R intervals and BPM in healthy adults. Further research is needed to determine if HRV measures with the Apple Watch offer a significant opportunity for the surveillance of CVD risk.

## 1. Introduction

Heart rate variability (HRV) represents the fluctuation in time intervals between consecutive heartbeats (R-R intervals) and serves as a vital physiological biomarker for the early detection of cardiovascular diseases [1,2]. These beat-to-beat variations, which can be measured in milliseconds, reflect subtle changes in the timing between heartbeats. For example, if there were 926 milliseconds between one heartbeat and the next, followed by 954 milliseconds between the second and third beat, the variability would be 28 milliseconds. These variations stem from complex interactions between multiple physiological mechanisms regulating heart rate (HR). The process begins at the sino-atrial (SA) node, often called the heart’s natural pacemaker, where specialized cells generate the electrical impulses that trigger heart muscle contractions [1,3].

The autonomic nervous system (ANS), which transmits signals from the central nervous system (CNS) to peripheral organs, maintains continuous control over the SA node’s activity [4]. This control system extends beyond heart rate regulation, influencing crucial functions such as blood vessel constriction/dilation and smooth muscle activity throughout the body [1,5,6]. The ANS operates through two complementary branches: the parasympathetic nervous system (PNS) and sympathetic nervous system (SNS). The PNS, responsible for “rest and digest” functions, promotes energy conservation and restoration by decreasing heart rate and increasing HRV [1,7]. Conversely, the SNS activates the “fight or flight” response during physical or psychological challenges, increasing heart rate while typically decreasing HRV [1,8]. The dynamic balance between these systems continuously adjusts in response to both internal and external stimuli, making HRV a sensitive indicator of the body’s adaptive capacity and stress response.

Overall, HRV reflects the collective impact of SNS and PNS activity on HR, serving as a measurable indicator of cardiovascular (CV) health and disease prognosis [1,9]. Analysis of HRV is useful in assessing overall cardiac health and regulation of the ANS. High HRV is associated with lower cardiovascular disease (CVD) incidence [10], lower mortality risk [11], and lower risk for dementia and cognitive decline [12], all indicators of healthy aging. Inversely, low HRV predicts CVD incidence and mortality in older adults, independent of traditional risk factors, such as left ventricular size and function [10,11]. Furthermore, experimental evidence suggests that HRV increases in a dose–response manner after as few as 6 months of exercise training [13]. Taken together, these findings suggest that HRV can be conceptualized as a modifiable factor that can reflect overall cardiovascular health and healthy aging.

To date, large-scale measurement of HRV has been challenging. HRV analysis has been conventionally performed in a laboratory environment on inter-beat intervals gathered from R waves in electrocardiograms (ECG) [14]. In the free-living environment, this measurement is typically acquired using an expensive and activity-limiting Holter monitor. However, recent advances in technology have resulted in the proliferation of low-cost, non-invasive wearable devices that typically include photoplethysmography (PPG) sensors that use a light source and a photodetector at the surface of the skin to measure volumetric changes in peripheral blood circulation that are the result of the heart beating [15]. Thus, these devices can be used to indirectly measure multiple components of the physiology of the heart, including HR at rest or during various levels of physical effort (including achieving maximal HR). Furthermore, recent validation studies have shown promising results of utilizing PPG to measure HRV in particular, though they note certain limitations such as data gaps and the need for controlled conditions [15,16].

Given that PPG sensors have been incorporated into widely available wrist-worn devices, such as the Apple Watch, there is now a possibility for widespread and periodic measurement of HRV in large numbers of individuals. At present, Apple Watch sales have surpassed 100 million [17] and represent nearly 40% of the smartwatch market share [18]. Despite the ubiquity of the Apple Watch, there is a shortage of independent research studies examining its validity for measuring HRV. Recent studies have shown varying results, as follows: while some demonstrate good reliability with reference devices for R-R intervals [15,16], others indicate potential underestimation of HRV measurements compared to chest-strap devices [19]. These discrepancies may be influenced by measurement timing and environmental conditions, particularly when comparing controlled laboratory settings to daily life measurements [19]. One laboratory-based study on 20 healthy volunteers (the age and sex of the participants were not reported) showed promising results, with 90% concordance between HRV measured via the Apple Watch and the Polar H7 chest belt [15]. This study was limited by a small sample size and a lack of demographic or anthropometric characterization of the study sample. Additionally, it did not interrogate various conditions under which HRV measurements with the Apple Watch are likely to occur (e.g., seated at a desk while working on a computer).

In the present study, we assessed the test validity of the Apple Watch’s measure of HRV by comparing it with HRV measured via Biopac 3-lead ECG, a research-grade device, among 78 healthy adults. We investigated measures taken under various controlled conditions in a laboratory under the direct supervision of trained staff. This represents an important step towards making an informed decision about whether the Apple Watch’s measure of HRV could be used within clinical practice and epidemiological research. Given that HRV is increasingly being used in cardiovascular disease risk stratification and, as it is a marker of overall cardiac health, the potential to measure it accurately, cheaply, and conveniently via a consumer-level wearable in a free-living environment has important implications for its widespread adoption.

## 2. Materials and Methods

### 2.1. Participants

Potential participants were recruited by word of mouth and via listservs (i.e., email). Research staff screened potential participants, and they were excluded if they were pregnant, had diabetes, or had any heart, lung, or autoimmune disease. Eligible participants were between the ages of 20 to 50 and 70 to 75 years old and were willing to abstain from alcohol for 72 h and from caffeine and nicotine for 24 h before testing. This sampling strategy was designed to capture both young/middle-aged adults and older adults, providing data points that allow for extrapolation across the age spectrum, including the intermediate age range. We recruited an approximately equal ratio of male and female participants, as well as an equal number of participants from each age group (~20 per decade).

### 2.2. Procedures

All study procedures were approved by the University of California, San Diego Institutional Review Board (approval number 200085). The participants provided written informed consent and attended one 2 h laboratory visit at the Exercise and Physical Activity Resource Center (EPARC) at UC San Diego.

Upon arrival, the participants were asked to complete four self-administered questionnaires covering socio-demographics, health history, smoking habits, medication use, physical activity, sedentary behaviors, and sleep patterns. EPARC staff then conducted physical measurements. Blood pressure (systolic blood pressure (SBP)/diastolic blood pressure (DBP)) and heart rate (BPM) were measured using an automatic monitor. Height and weight were measured twice without shoes using a stadiometer (to nearest 0.1 cm) and calibrated digital scale (Seca, Chino, CA, USA; to nearest 0.1 kg). For hip and waist circumference measurements, participants stood on a 12″ box with their feet together, weight evenly distributed, and arms across their chest. Using Gulick II Tape (Country Technology, Inc., Gays Mills, WI, USA), staff measured the maximum buttocks extension and narrowest torso portion twice (to nearest 0.1 cm).

The participants were instrumented with the Biopac 3-lead ECG (BIOPAC Systems, Inc., Goleta, CA, USA), the criterion device, and the Apple Watch (Apple, Cupertino, CA, USA). The Biopac 3-lead ECG was selected as the reference standard for measuring HRV. The Biopac system sampled ECG data at 2000 Hz to ensure accurate R-wave detection. Electrodes were placed in a chest-mounted pattern with one electrode under each clavicle and the third on the lower left rib cage. The Apple Watch was placed on the participants’ left wrist. The proper fit of the Apple Watch required skin contact on the top of the wrist, with the electrical and optical heart sensors being snug but comfortable, with room for the skin to breathe to ensure adequate readings. The watch utilizes PPG technology to record HRV, using green LED lights and light-sensitive photodiodes to detect blood flow in the wrist. While the Apple Watch flashes its LED lights hundreds of times per second for heart rate detection, the exact sampling frequency for HRV measurement varies based on the measurement context, with higher sampling rates during active measurements such as during Breathe sessions [20].

We utilized the Breathe app on the watch, as it is the current method used to ensure a real-time HRV measure. The Breathe app allows measures to last between one and five minutes, with the number of breaths per minute ranging from four to ten. However, we found that the actual Breathe app is twelve seconds longer than the whole minute chosen (i.e., a five-minute session would be five minutes and twelve seconds). For our study, we programmed the Breathe app to complete seven breaths per minute and chose either five- or one-minute durations based upon activity (see Table 1 below). While the app provided visual breathing cues, the participants were not specifically instructed to follow app commands, but rather encouraged to breathe at their own pace. We turned off the haptic vibrations so as to not distract the participants. A synchronized countdown was conducted for each condition to ensure that the Apple Watch and Biopac 3-lead ECG recordings aligned. For that, we used event markers electronically placed on the Biopac 3-lead ECG recording when the Apple Watch Breathe app began and ended.

HRV was measured under the following conditions during the visit: (1) at rest in a supine position, (2) sitting while talking, (3) sitting while watching a movie clip, (4) sitting before walking, and (5) after walking. Due to the sampling capabilities of the Apple Watch, five minutes of HRV was measured for the first three conditions and two one-minute segments for the last condition. The Biopac 3-lead ECG continuously provided continuous measurement throughout each condition. In order to calculate HRV, we first derived R-R intervals based on the time (in ms) between R waves detected on the ECG. We then calculated an n value by calculating the absolute value of the difference in the RR interval for the beat in question from the RR interval for the previous beat. N-N intervals were then calculated with the absolute value of the difference in the n value for the beat in question from the n value for the previous beat. MeanRR was estimated from BioPac’s single lead EKC, which provides a uniformly sampled RR time derived from a 2000 Hz sampling rate, and, in order to estimate meanHR, we utilized BioPac’s software (AcqKnowledge 6.0.0) to remove presumed movement-based artifact and re-interpolated meanHR.

For the Apple Watch measurements, raw data from the Apple Health app were extracted using the “Export All Health Data” function. The data were converted to text format, and pulse rate variability (PRV) based on the timestamp data provided by Apple (labeled as HeartRateVariabilityMetadataList) was used to estimate RR intervals and was extracted at the level of centiseconds. In order to acknowledge that photoplethysmography sensors in the Apple Watch technically measure PRV rather than true RR intervals derived from ECG, we used the terminology “PRV-estimated RR data” to identify the RR (and derived variables) context of heart rate variability analysis. This extraction process allowed us to analyze the temporal patterns at a granular level for comparison with our reference measurements. In the event that Apple was unable to provide a uniformed sampling value, the corresponding BioPac data were not included in the analysis.

For the first condition, the participant lay supine in a quiet room for 10 min. Immediately following the 10 min, the Apple Watch and Biopac 3-lead ECG gathered data for five minutes. The talking condition was then conducted with the participant being seated and data being gathered for the full five-minute conversation. A script of questions was used as a reference to keep the participants engaged in conversation during the measurement period. While remaining seated, a five-minute clip from the film Harry Potter was shown for participants aged 20 to 50 years, and a clip from the film Indiana Jones was shown for participants aged 70 to 75 years. Data were collected continuously for five minutes. Lastly, HRV data were collected while the participants were seated with the Apple Watch and Biopac 3-lead ECG for one minute immediately prior to, and one minute following, the 2.5-min walk.

For each Apple Watch measurement, the compatible iPhone was examined in real-time to ensure that the watch recorded the data properly. If the watch did not record HRV, another attempt was conducted for the condition for a maximum of two attempts per condition. In the event that Apple was unable to provide a uniformed sampling value, the corresponding BioPac data were not included in the analysis. The key metrics, as shown in Table 2, included the average and standard deviation of PRV-estimated R-R intervals (in ms), the average and standard deviation of N-N intervals (in ms) between consecutive valid PRV-estimated R-R measurements, and average beats per minute. Biopac 3-lead ECG data were downloaded using AcqKnowledge 4.4 software, where each cycle was found and copied into the same excel database. Event markers on the Biopac 3-lead ECG, which were flagged during each of the four conditions at the beginning and end of the Breathe session, were also copied into the excel database, serving as an alignment tool for the Biopac 3-lead ECG and Apple Watch. A final summary table comparing the events of the Biopac 3-lead ECG with those of the Apple Watch was generated from the excel database analyzing matched beats and PRV-estimated R-R differences. All data were associated with a unique participant identifier (e.g., 00001) and downloaded and stored on secure, password-protected EPARC servers.

### 2.3. Statistical Analysis

Demographic and anthropometric characteristics were analyzed using univariate descriptive statistics. We evaluated test validity using the Bland–Altman method to analyze the agreement between the Apple Watch and the reference device. The limits of agreement were specified as the mean of the discrepancies ±1.96 times the standard error of the differences. The corresponding Bland–Altman plot visualized how agreeable the devices were represented by the mean of the differences by the means of the measures, with the limits of agreement. No formal statistical tests were performed to assess systematic bias in the Bland–Altman analysis. Bias was evaluated using mean differences and 95% confidence intervals (CIs) for agreement and visual assessment of Bland–Altman plots to check for patterns indicating proportional bias. All performance metrics, including MAPE, were calculated at the minute level, rather than on features averaged over the entire duration of each condition. The mean absolute percentage error (MAPE) was computed as the average of absolute differences among the measures, divided by the applicable research-grade measure, multiplied by 100. The combination of both precision and accuracy was estimated using Lin’s correlation coefficient, which was visualized on a scatter plot. The aforementioned metrics were calculated for each measure and condition of interest. Outliers were identified as any observation that exceeded 3 standard deviations from the mean of the difference. After removing the outliers, we further analyzed the data using the same statistical analysis capturing approximately 99.7% of the observations. For comprehensive analysis, we generated Bland–Altman plots and scatter plots with Lin’s correlation coefficients for each metric under all conditions, analyzing the data both with and without outliers. These plots are available in the Appendix A.

## 3. Results

From February 2021 to August 2021, 78 participants (34 males and 44 females) completed the study and were included in the data analyses. The mean (SD) age was 43.6 (18.8) yr, height was 171 (9.38 cm), weight was 72.5 (14.8) kg, and BMI was 24.7 (4.02) kg/m^2^ (Table 3).

As shown in Table 4, the analysis of Apple Watch accuracy against a 3-lead ECG revealed varying performance across conditions. For PRV-estimated R-R intervals, the mean bias ranged from −1.67 to −32.84 ms, with MAPE values between 1.15% and 5.66%. N-N intervals showed greater variability, with the mean bias ranging from 0.48 to 18.93 ms, and notably higher MAPE values (31.41–93.08%). BPM measurements demonstrated the most consistent performance, with mean bias between 0.37 and 3.45, and MAPE values from 1.16% to 6.46%.

Table 5 demonstrates that device data capture reliability varied substantially across conditions, with failure rates ranging from 2.56% in Condition 1 to 43.59% in Condition 2, indicating environment- or activity-dependent reliability.

### 3.1. PRV E Intervals

Figure A1 presents Bland–Altman and Lin’s correlation coefficient analyses with corresponding plots for PRV-estimated R-R intervals for Condition 1, including and excluding outliers. Plots for the remaining conditions can be found in the Appendix A. Outliers were defined as observations exceeding ± 3 standard deviations from the mean difference in each condition. The pre- and post-walking conditions showed the largest measurement disagreement, with a MAPE of 4.59% and 5.66%, respectivley. The resting condition demonstrated the smallest disagreement, with a MAPE of 1.15%.

### 3.2. N-N Intervals

Figure A2 presents Bland–Altman and Lin’s correlation coefficient analyses with corresponding plots for N-N intervals for Condition 1, excluding outliers. Plots for the remaining conditions can be found in the Appendix A. The resting condition demonstrated the smallest measurement disagreement, with a MAPE of 31.41%, while the conversation condition showed the largest disagreement, with a MAPE of 93.03%.

### 3.3. BPM

Figure A3 presents Bland–Altman and Lin’s correlation coefficient analyses, with corresponding plots for Condition 1, for BPM, excluding outliers. Plots for the remaining conditions, including and excluding outliers, can be found in the Appendix A. The post-walk condition demonstrated the largest disagreement, with a MAPE of 6.46%, while the resting condition showed the smallest disagreement, with a MAPE of 1.15%.

## 4. Discussion

This study evaluated the validity of the Apple Watch’s HRV measurements against a research-grade Biopac 3-lead ECG in healthy adults. We assessed the performance across conditions designed to simulate daily activities, including resting, conversation, watching a film, and pre/post walking. When comparing HRV measures between devices, we found the highest accuracy for PRV-estimated R-R intervals and BPM across all conditions.

We found strong agreement between devices for PRV-estimated R-R intervals and BPM, with a MAPE below 6% and 7%, respectively, across all conditions. These results align with previous Apple Watch validation studies, particularly during resting conditions, where we observed optimal performance [15]. The resting state’s superior accuracy is particularly relevant given that the Apple Watch’s Breathe app is designed for use during stationary, quiet periods [15]. However, it is important to note that other researchers have found an underestimation of Apple’s HRV measurements when gathered in vivo across several days of observation [19]. These discrepancies may be influenced by measurement timing and environmental conditions, particularly when comparing controlled laboratory settings with daily life measurements [19]. Additionally, differences in protocol may introduce artifacts such as movement or temperature changes that could introduce artifact or affect peripheral vessel compliance in ways that would affect PRV but not electrical estimations of R peaks. Similarly, differences in the Apple Watch model or associated software used in the various studies may explain the divergence in results. Finally, utilizing the Breathe app to induce PRV-based recordings of the heart rhythm rather than passive HRV measurements gathered “automatically” by the Apple device, or gathered using electrical signal via the ECG function, may have also influenced our accuracy results compared to those of previous studies. Although the participants were not required to strictly follow the Breathe app hepatic/visual prompts, the presence of respiratory cues may have induced changes in breathing, which could impact HRV results.

Since HRV measurements are commonly taken while resting, high agreement was expected for the resting condition, which produced the most favorable results. The Breathe app was created to remind individuals to take time to breathe each day, guiding them through a sequence of deep breaths. The resting condition would most closely mimic the atmosphere in which the Breathe app was intended to be used, since most breathing sessions suggest finding a quiet place to sit or lie down and focus solely on breathing, limiting movement [21]. Therefore, the resting condition would minimize any additional movement artifact, allowing the body to relax and the watch to detect blood flow more accurately on the wrist.

The analysis of N-N intervals revealed the most significant discrepancy, with an average MAPE of 68%—approximately 20 times higher than that of the PRV-estimated R-R intervals and BPM. The largest disagreements occurred during conversation (MAPE 93.08%), pre-walking (85.25%), and post-walking (83.77%) conditions. We identified several potential mechanisms for these discrepancies, including hand gestures during conversation likely causing watch movement, changes in respiratory patterns during speech that may have affected blood flow detection, and increased sympathetic tone following physical activity potentially impacting measurement accuracy. While this suggests poor agreement, the small absolute values of the N-N intervals translate into large percentage variations. Therefore, mean error provides a more meaningful metric for assessing agreement in this context.

Our outlier analysis revealed a striking age-related pattern in measurement accuracy. Among the participants aged 70–75 years, we observed a disproportionate number of outliers across all metrics, as follows: 73% of PRV-estimated R-R interval outliers, 63% of BPM outliers, and 84% of N-N interval outliers. Most notably, a single 73-year-old participant accounted for 68% of all N-N interval outliers. This clustering of outliers in older adults suggests that age-related changes in skin properties, such as reduced perfusion, altered melanin content, or decreased dermal thickness, may significantly influence the Apple Watch’s measurement accuracy. These findings highlight an important limitation of photoplethysmography-based wearable devices and emphasize the need to investigate how age-related skin changes affect sensor performance. While age itself may represent a potentially influential factor in PPG accuracy, we were not able to investigate how skin changes directly impact the results. Our recruitment intentionally spanned a wide age range (20–75 years) in order to capture potential physiological variation due to age. Importantly, age is used as a factor within the proprietary algorithms of consumer-level devices, yet its influence on accuracy is unclear. This concern extends to skin tone variation as well—a systematic review across 10 studies with 469 participants showed that evidence of reduced accuracy in darker-skinned individuals remains inconclusive, with studies showing mixed results. However, these findings suggest potential measurement bias that warrants further investigation with larger, more rigorously designed studies [22].

Additionally, our analysis revealed patterns in data quality issues that advance our understanding of the device’s limitations. The Apple Watch frequently failed to record measurements during non-standard use conditions, which we interpret as a built-in safety feature preventing inaccurate data collection. Additionally, we observed that poor sensor contact and varying skin perfusion may have contributed to missing data points, consistent with previous studies documenting intermittent measurement gaps [15,16].

With the promising results of BPM and PRV-estimated R-R intervals of this study, Apple Watches potentially offer a significant opportunity for the surveillance of CVD and the development of interventions to reduce CVD risk. This is important because the population of older Americans is projected to nearly double from 49 million to 95 million by 2060, owing partly to declines in CVD mortality over the past half-century [23]. Successful medical and public health interventions have focused primarily on preventing CVD events, however, as CVD remains the leading cause of death in the US, novel intervention targets are needed to promote healthy cardiovascular aging. HRV and the accurate prescription of relative exercise intensity are two potential targets. Having access to a continuously monitored HR could offer significant opportunities to better understand the role that HRV plays in long-term CV health, offering new insights into population health. However, it is important to note that our study only validated measurement accuracy in a controlled setting. While these findings represent a step toward understanding wearable technology capabilities, further research is necessary to further understand clinical relevance and whether Apple Watches can be effectively used for CVD surveillance or for the development of interventions to reduce CVD risk. Having access to wearable sensors to help monitor CV health could allow population health to progress steadily with technology in areas such as stress management, athletic performances, and clinical care.

This study adds to the existing literature in this rapidly expanding field of study. In general, our findings were in agreement with two systematic reviews examining 22 studies with 714 participants, which found that wearable devices (not exclusive to Apple) showed excellent validity for HRV measurements when the measurements were gathered with participants at rest, with intraclass correlation coefficients (ICCs) ranging from 0.85 to 1.00 compared to ECG recordings [24]. However, measurement accuracy significantly deteriorated during physical exercise or when motion was introduced, with ICCs decreasing as activity levels increased [25]. Our findings agree with these pooled results with generally high levels of agreement with ECG-based HRV calculations during rest and decreasing levels of agreement as movement increased.

Our study had some limitations. One major limitation was the reliability of the Apple Watch’s HRV recording while using the Breathe app. The limited recordings could have been a result of the Apple Watch positioning, not having good skin contact on the wrist, movement of the arm and wrist, or could have been affected by the participants’ skin perfusion. This observation is consistent with other studies that have documented data gaps in Apple Watch HRV measurements, averaging five gaps per recording with lengths of 6.5 s [16]. While these gaps significantly affect frequency domain metrics, time-domain HRV indices remain relatively unaffected [15,16]. Further research is needed to determine the optimal position of watch placement on the wrist for the successful assessment and reliability of the watch. Another important limitation relates to the specific HRV metrics available for analysis. While we were able to calculate basic time-domain features such as sdNN and RMSSD, the Apple Watch does not provide access to the data needed to investigate the accuracy of high-frequency band power, which is considered one of the most valuable parameters in HRV analysis for assessing parasympathetic activity. Furthermore, our findings may not be generalizable to the free-living context, since they were associated with a laboratory-based protocol. More research is needed for the Apple Watch’s HRV measure in a free-living context to determine its validity for use in interventions, epidemiologic studies, or personal use. Lastly, since our sample was a convenient, non-clinical population, our results may not be generalizable to other populations. For this reason, further research on the Apple Watch in multiple populations is required.

On the other hand, our study included numerous strengths. This was one of the first studies to objectively measure HRV using two devices, one being a wrist-worn device. Our protocol included a variety of controlled conditions, exposing how the Apple Watch operates under different circumstances. These conditions were similar to the conditions of daily living that an individual would consider when assessing their HRV. Lastly, unlike other Apple Watch validity studies, our study included stratification across different age decade groups, which improves the generalizability of our results across a variety of ages.

## 5. Conclusions

A valid, economical product that accurately measures HRV among young adults is important to further cardiovascular health regarding risk stratification and disease identification. This is among a small number of studies to evaluate the validity of the Apple Watch’s definition of HRV against a reference device. The high level of validity for measuring BPM and PRV-estimated R-R intervals on the Apple Watch provides strong implications for clinical and epidemiological research. With the general population having access to an accurate wrist-worn health device, new interventions could target real-time applications to lower CVD. The next step would be to test validity in a free-living environment to determine if the Apple Watch’s HRV measures offer a significant opportunity to surveil CVD risk.

## Figures and Tables

**Table 1 sensors-25-02380-t001:** Laboratory test conducted for comparison of Apple Watch and Biopac 3-lead ECG Device.

Category	Activity	Event	Time	Description
Lying Down	Rest	1	5 min	Lying quietly
Sitting	Conversation	2	5 min	Engaging in a conversation
Sitting	Video Clip	3	5 min	Watching a movie clip
Standing	Pre-Walk	4	1 min	Standing quietly before the 2.5-minute walk
Standing	Post-Walk	5	1 min	Standing quietly after the 2.5-minute walk

**Table 2 sensors-25-02380-t002:** Data dictionary defining variables of interest for HRV measurement.

Variable of Interest	Definition
PRV-estimated r-r_average	Average length of time of valid PRV-estimated R-R (ms)
PRV-estimated r-r_sd	Standard deviation of the length of valid PRV-estimated R-R (ms)
n-n_average	Average length of valid N-N (ms), defined as the length of time between two consecutive valid PRV-estimated R-R measurements
n-n_sd	Standard deviation of the length of valid N-N (ms)
Avg_bpm	Average beats per minute

**Table 3 sensors-25-02380-t003:** Characteristics of included participants (n = 78) described as mean (SDs) or frequencies and (%).

N	78 (100)
Age (yr)	
20–29	21 (26.9)
30–39	20 (25.6)
40–49	17 (21.8)
70–75	20 (25.6)
Sex (%)	
Female	44 (56.4)
Male	34 (43.6)
Hispanic Origin (%)	10 (12.8)
Ethnicity (%)	
White	55 (70.5)
Black	1 (1.28)
Asian	17 (21.8)
American Indian	1 (1.28)
Native Hawaiian	1 (1.28)
Other	3 (3.85)
Height (cm)	171 (9.38)
Weight (kg)	72.5 (14.8)
BMI (kg/m^2^)	24.7 (4.02)

**Table 4 sensors-25-02380-t004:** The number of times the Apple Watch failed to capture data at the minute level, described as N (78) and frequencies (%).

N	78 (100)
Condition 1	2 (2.56)
Condition 2	34 (43.59)
Condition 3	10 (12.82)
Condition 4	21 (26.92)
Condition 5	18 (23.08)

**Table 5 sensors-25-02380-t005:** Agreement analysis between Apple Watch and Biopac 3-lead ECG measurements across different experimental conditions.

	n	Mean Bias	SD	MAPE (%)	Lin’s CCC
PRV-estimated R-R Average (ms)					
Condition 1	368	−1.67	20.77	1.15	0.991
Condition 2	192	−5.83	45.05	3.40	0.961
Condition 3	313	−12.79	32.58	2.36	0.973
Condition 4	55	−26.69	75.86	4.59	0.803
Condition 5	57	−32.84	69.70	5.66	0.876
N-N Average (ms)					
Condition 1	371	3.11	21.33	31.41	0.737
Condition 2	176	11.98	40.87	93.08	0.044
Condition 3	308	0.48	27.78	45.43	0.476
Condition 4	55	12.92	46.43	85.25	0.323
Condition 5	57	18.93	38.84	83.77	0.394
BPM Average (BPM)					
Condition 1	368	0.37	1.02	1.16	0.993
Condition 2	205	1.45	3.26	3.22	0.954
Condition 3	325	1.02	1.70	2.30	0.979
Condition 4	55	2.54	3.96	4.88	0.879
Condition 5	58	3.45	4.48	6.46	0.874

n = Number of valid observations after outlier removal. Mean bias = Average difference between Apple Watch and Biopac measurements. SD = Standard deviation of the differences. MAPE (%) = Mean absolute percentage error. Lin’s CCC = Lin’s concordance correlation coefficient (ranges from −1 to 1, where 1 indicates perfect agreement).

## Data Availability

The data presented in this study are available upon reasonable request from the corresponding or senior author.

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
