# Peer review of "Validity of Heart Rate Variability Measured with Apple Watch Series 6 Compared to Laboratory Measures"

_sensors, 2025, doi:10.3390/s25082380_

Round 1
Reviewer 1 Report
Comments and Suggestions for Authors
In this paper, authors investigate the performance of Apple Watch PPG sensor to estimate HRV-related basic features (i.e., meanRR, meanNN, and meanHR) compared to a well-know research-grade device. Results align with the rest of the literature comparing HRV estimates obtained from wearable PPG devices and ECG, respectively.
Comments:
1) As also stated by the authors, age plays a confunding role on the PPG estimates. However, they did not take it into account in their analyses. The effect of age should be included in the analyses.
2) lines 132-139: authors should clearly indicate the sampling frequency of the smartwatch, as well as the one adopted to acquire the ECG using Biopac.
3) Table 2 is never cited in the text. Moreover, a section explaining how HRV time series were estimated from the biopac device and which features were extracted is missing. Did the authors estimate meanRR from the non-uniformely sampled RR time series? And was meanHR estimated from the re-interpolated HRV time series? Also, a brief description of these features would be necessary. It is also not specified how the authors derived HRV time series from ECG.
4) The statistical analysis section should be enriched with details about statistical tests carried out, as well as corrections and adjustments for multiple comparisons. In particular, did the authors test for the statistical significance of biases in the BA analysis? Relevant information is missing.
5) It is not clear from the text if performance measurements (e.g., MAPE) have been computed on the features averaged over the entire duration of each condition
6) The clarity of the caption of table 5, including the legend, should be improved.
7) Figures quality should be improved, as they result a bit blurred. Also, captions should provide a more detailed description so that figures could be better understood.
8) Authors state that this is one of the first studies comparing HRV measures between a wearable PPG device and a research-grade PPG device. However, only meanRR/meanNN and meanHR are considered. Authors should at least investigate the accuracy of other more specific time-domain features of HRV, such as sdNN and RMSSD. The latter is know to be correlated with parasympathetic activity, and would add novel and interesting insights to this study. Besides time-domain HRV measures, authors could try to investigate the accuracy of high-frequency (HF) band power, which is perhaps one of the most interesting parameters in HRV analysis.
Reviewer 2 Report
Comments and Suggestions for Authors
General Comments
This is a timely study with a clear research question, but I have serious concerns about the methodology and the way the findings are interpreted. Before I can recommend publication, the authors need to clarify key aspects of their methodology, address discrepancies in their reported metrics, and adjust their conclusions to reflect what was actually measured.
The biggest issue is a fundamental disconnect between the reported "Metrics of Interest" and how HRV is actually measured using the Apple Watch. The authors claim to have extracted RR intervals (R-R and N-N) from the Apple Watch's Breathe app, but this is not possible using PPG technology. The Apple Watch does not record ECG data when using the Breathe app—it estimates pulse rate variability (PRV), which is derived from changes in blood volume, not direct cardiac electrical activity. This means the authors may have either: (1) misunderstood the nature of their own data, or (2) misreported the methodology.
Adding to this issue, the paper does not state which Apple Watch model was used. This is a major omission. Different Apple Watch generations have different PPG sensor hardware and signal processing algorithms, which directly affect HRV measurement accuracy. If multiple Apple Watch models were used, this would introduce uncontrolled variabilityin the results. If the study used only one model, that model should be explicitly stated.
Beyond these issues, the authors overstate the implications of their findings. Their study only validates the Apple Watch under highly controlled conditions using a manual HRV measurement (Breathe app). But in real-world use, Apple’s HRV data comes from passive, algorithm-driven measurements, not from manually triggered recordings. This distinction is critical. Their results do not support the Apple Watch as a tool for real-world CVD risk assessment, yet they suggest this in their conclusions. The authors should temper their claims and acknowledge that their findings only confirm pulse rate variability agreement under controlled conditions—not broader clinical utility.
Lastly, the discussion does not adequately compare these findings to prior Apple Watch HRV studies. Reference 20, for example, reported lower accuracy for Apple Watch HRV measurements, but the authors do not explore why their results differ. The differences in protocol, sample characteristics, or even which Apple Watch model was used could explain this discrepancy. The authors need to address this.
Specific Comments
Methods
Several aspects of the methodology are unclear or problematic.
First, the exclusion of participants aged 51-69 years old is unexplained. This is a highly unusual cutoff, particularly since HRV declines with age. The authors need to explain why this age group was omitted, as it may introduce bias into their findings.
Second, the Apple Watch model used in the study is not reported. This is a serious issue. HRV measurement accuracy can vary between different Apple Watch series due to changes in PPG sensor technology and processing algorithms. If multiple models were used, the authors need to justify why they did not control for this. If only one model was used, they must specify which one. Without this information, the reliability of their findings is questionable.
Another unclear point is the sampling method. The authors say, “Due to the sampling capabilities of the Apple Watch, five minutes of HRV were measured for the first three conditions and two one-minute segments for the last condition.” It is not clear why the task duration affected how the Apple Watch sampled HRV. Apple’s passive HRV measurement is algorithm-driven, and the Breathe app's timing should not be task-dependent. If the Breathe app enabled greater control over measurement intervals, the authors should explain how.
There is also a typo in “film Indian Jones.”
Data Collection and Metrics
The most significant problem is the claim that "Raw RR data was extracted from the Breathe app and stored in the iPhone’s Personal Health Record."
This is not possible using PPG alone. The Apple Watch only records PRV (pulse rate variability), not true RR intervals, when using the Breathe app. RR intervals can only be obtained through an ECG, which the Apple Watch does have—but the authors never mention using the ECG function.
If they used PPG-derived PRV, then they should not report RR or N-N intervals in their results. Instead, they should use the correct terminology: pulse interval variability (PIV).
If they used the Apple Watch’s ECG function, they must clearly state this in the methods. Right now, there is a fundamental mismatch between what the Apple Watch can measure and what the authors claim to have analyzed.
This issue extends to Table 2 ("Metrics of Interest"). None of the variables listed—R-R average, N-N average, etc.—can be directly measured through PPG. The authors need to either revise the table to reflect PRV data or explain how they obtained these metrics. If they incorrectly assumed that PRV = RR intervals, this is a major methodological flaw.
Results
Because of the methodological issues above, the results are difficult to interpret. If the Apple Watch only measured PRV, then the reported HRV metrics (RR intervals, N-N intervals) are incorrect. If the Apple Watch’s ECG function was used, the authors need to state this explicitly. Right now, the results do not align with the reported methodology.
The comparison with previous studies is also weak. Reference 20 reported lower accuracy for Apple Watch HRV measurements, but the authors do not explain why their findings differ. Possible explanations include different Apple Watch models, different measurement conditions, or differences in data handling. The authors should address this.
Until the authors clarify these issues, it is difficult to trust that their findings are accurate or meaningful. At minimum, they need to reframe their results as a validation of PRV measurement, not HRV in the traditional sense. If they fail to do this, their conclusions will be misleading.
As it stands, I cannot recommend publication until these concerns are fully addressed.
Round 2
Reviewer 1 Report
Comments and Suggestions for Authors
The authors addressed all my concerns.
Reviewer 2 Report
Comments and Suggestions for Authors
General comments:
The revised manuscript still has serious problems. The authors have acknowledged that the Apple Watch measures pulse rate variability (PRV) rather than true heart rate variability (HRV), but they still misrepresent what was actually measured (they claim to have extracted R-R intervals from the Apple Watch’s Breathe app, but this is not possible using PPG). The Apple Watch does not provide raw R-R intervals unless the ECG function is used, which they did not do. This needs to be corrected.
Beyond this, the discussion section is incomplete. The authors ignore previous studies that found lower accuracy for Apple Watch HRV measurements and do not explain why their results differ. They also fail to cite a recent umbrella review that summarizes wearable HRV accuracy across multiple studies. This makes their conclusions weaker.
Finally, the paper overstates the Apple Watch’s potential for CVD risk assessment. The study only validated PRV under controlled conditions, but the authors suggest it has clinical utility. This is misleading. They need to clearly state what their findings actually show and avoid making claims beyond what their data support.
These issues must be fixed before the paper can be considered for publication.
Specific comments
Thank you for your response and for acknowledging the distinction between PRV and true HRV derived from R-R intervals in ECG recordings. However, the revisions you have made still contain significant inaccuracies and misrepresentations of what the Apple Watch is capable of measuring.
You state: "While we acknowledge that photoplethysmography sensors in the Apple Watch technically measure pulse rate variability (PRV) rather than true RR intervals derived from ECG, we used the terminology 'RR data' as it is commonly applied in the broader context of heart rate variability analysis."
This is misleading and incorrect. While it is true that PRV is often used as a proxy for HRV, PRV is not the same as RR intervals, and using "RR data" in this context is methodologically inaccurate. PRV is derived from pulse wave intervals detected via PPG, which can be influenced by changes in vascular tone, respiratory rate, and motion artifacts, making it not equivalent to ECG-derived R-R intervals.
You must explicitly state that your measurements are based on PRV, not RR intervals. Using the term "RR data" is scientifically incorrect unless your study included ECG-derived HRV from the Apple Watch’s ECG function (which it did not).
Furthermore, regarding your claim that Apple Watch provides raw RR Intervals in text format, you state:
"For the Apple Watch measurements, raw data from Apple was converted to text format, and RR intervals were then extracted at the level of the centisecond."
I beleive this is factually incorrect. The Apple Watch’s Breathe app does not provide raw RR intervals, nor does it allow for the extraction of centisecond-level beat-to-beat intervals. Instead, Apple only provides summary HRV metrics (e.g., SDNN) through HealthKit, which are derived from PRV estimates. There is no way to extract true RR intervals from Apple’s Health data unless the ECG function is used.
If you are claiming to have obtained RR intervals, you must explicitly clarify: how you accessed this data, which Apple Health API or third-party tool was used and whether the data represents PPG-derived PRV intervals rather than ECG-derived RR intervals
Finally, regarding the following: "In the event that Apple was unable to provide a uniformed sampling value, the corresponding BioPac data was not included in the analysis."
This suggests that you expected Apple’s HRV measurements to be uniformly sampled, which is not how Apple processes HRV data. My understanding is that the Apple Watch’s Breathe app does not provide a continuously sampled RR interval series like an ECG; it instead computes summary statistics at pre-defined time points. There is no “uniform sampling” in Apple’s PRV data.
Because the methodology still misrepresents what was actually measured, this raises concerns about the validity of comparisons made between the Apple Watch and the Biopac ECG, the accuracy of HRV-derived metrics reported in the paper and the conclusions drawn about the Apple Watch’s ability to measure HRV.
The Apple Watch data should be properly reframed as PRV-derived metrics, and any analyses or conclusions based on "RR intervals" should be revised to reflect this. Otherwise, the study overstates the Apple Watch’s capability and risks misleading readers about its accuracy for HRV measurement.
Discussion
Next, the discussion fails to adequately contextualize the findings within the broader body of research. In my original review of this paper, I noted the need to explain discrepancies between their findings and Reference 20 (O’Grady et al), which reported lower accuracy for Apple Watch HRV measurements. You seem to have ignored this point. In the revised version, you should explore how differences in protocol (e.g., measurement conditions, Apple Watch model used) and analysis approach might explain the divergence. You also do not acknowledge whether your methodology (e.g., manual Breathe app recordings vs. passive HRV readings) may have influenced your accuracy results compared to previous studies.
Beyond this, you state that: "With the promising results of BPM and R-R intervals of this study, Apple Watches potentially offer a significant opportunity for the surveillance of CVD and the development of interventions to reduce CVD risk."
This is overly ambitious given the methodological limitations of your study. You only validated manually triggered PRV measurements under controlled conditions, not real-world, algorithm-driven CVD risk assessment. HRV alone is not a clinically validated tool for CVD risk stratification. While it has potential, no major guidelines endorse it as a standalone CVD risk marker. You also make no attempt to reconcile your conclusions with the limitations of PPG-based PRV, which is significantly influenced by motion artifacts, vascular changes, and skin tone variations. In the revised version, you need to temper your conclusions. You have only validated PRV measurement under controlled conditions, not its clinical relevance—you need to clearly differentiate between what your study shows (accuracy under controlled conditions) versus what you cannot claim (clinical utility for CVD surveillance).
More broadly, since this study was conducted, a comprehensive umbrella review on wearable HRV measurement accuracy (DOI: 10.1007/s40279-024-02077-2) has been published. This paper synthesizes all available evidence on HRV accuracy across different consumer wearables, including Apple Watch. The umbrella review likely resolves the very inconsistencies that you failed to discuss. It provides a higher-level evidence base that could strengthen AND challenge your conclusions. I would urge the authors to integrate the latest research into their discussion—which includes <10 citations to the wider literature. This field is evolving rapidly – and the authors need to better “place” their paper in the context of the field.
